# Formation of Nanowires of Various Types in the Process of Galvanic Deposition of Iron Group Metals into the Pores of a Track Membrane

**DOI:** 10.3390/membranes12020195

**Published:** 2022-02-08

**Authors:** Dmitri Zagorskiy, Ilia Doludenko, Olga Zhigalina, Dmitrii Khmelenin, Vladimir Kanevskiy

**Affiliations:** 1FSRC «Crystallography and Photonics» RAS, 119 333 Moscow, Russia; doludenko.i@yandex.ru (I.D.); zhigal@crys.ras.ru (O.Z.); xorrunn@gmail.com (D.K.); kanev@crys.ras.ru (V.K.); 2Department of Material Science, Bauman Moscow State Technical University, 105 005 Moscow, Russia

**Keywords:** nanowires, polymer membranes, template synthesis, microscopy, X-ray analysis

## Abstract

The processes of formation of one-dimensional nanostructures by the method of matrix synthesis was studied in this work. Nanowires (NWs) from magnetic metals of iron-group and copper (3-d metals) were synthesized in the pores of matrix-track membranes by galvanic deposition. NWs with both homogeneous elemental distribution (alloys) and with periodically alternating parts with different composition (layers) were obtained in matrices with different pore diameters and under different parameters of the galvanic process. The transport of ions, which determined the growth of wires, in pores of different sizes was analyzed. The influence of the size of pore channels on the features of NWs growth, the correlation between the elemental composition of the NWs and the growth electrolyte, as well as the influence of the growth conditions (voltage and pore diameter) were investigated. Approaches to formation of thin layers in layered NWs were studied. This included the choice of methods for controlling the pulse duration, slowing down the growth rate by the dilution of the solution, the use of additives and the work with reference electrode. The study of NWs was carried out using visualization and analysis of their structure using transmission and scanning electron microscopy, electron diffraction, energy dispersive analysis, and elemental mapping. For the studied types of samples, a relationship was established between the growth conditions and the structure. This data raises the possibility of varying the magnetic properties of NWs.

## 1. Introduction

Track membranes (TMs) are a special type of membrane. A TM is a thin polymer film with many identical through-pores. In general, TMs are a product of radiation technologies. It was found that when microparticles (ions, fission fragments) pass through some polymers, hidden (so-called “latent”) tracks are formed [1,2,3]. Subsequent visualization of these tracks made it possible to study parameters of microparticles. Later, this approach was used to obtain porous films (membranes). For this, particles with known parameters (for example, accelerated heavy ions), which uniformly irradiated the surface of a polymer film, were used. The irradiated film was then subjected to a special treatment—sensitization and etching of latent tracks. This approach made it possible to obtain a thin-film material with through-pores of the same size and controlled quantity per unit area (the so-called pore density). Most often they have a regular cylindrical shape and are directed perpendicular to the surface (respectively, parallel to each other).

The resulting porous membrane material, due to the uniformity of all pores, and the possibility of changing them within wide range, began to be used as filters (so-called nuclear filters or track membranes) [4]. From the filtration point of view, they are ideal sieves, since they have narrow pore size distribution and, accordingly, high selectivity. For some purposes, it is also important that the filtration takes place on the membrane surface. For about four decades, such materials have found practical application for fine filtration—in medicine, biology, and the food industry.

Another area of using TMs was a method developed in the 1990s, so-called “matrix synthesis” (or “template synthesis”) [5,6,7]. In this case, the pores in the film were used as nano-sized containers, which were filled with the required material. As a result, pore replicas were formed—as a rule, repeating their shape, i.e., in the form of elongated cylinders. These structures with a high aspect ratio (diameters in the tens of nanometers and lengths of several microns) are called nanowires (NWs, nanorods, nanoneedles). With this approach, they are formed inside the pores of TM, as a rule, in the form of an array (ensemble) of parallel oriented structures with a high density of arrangement in space. Since the mid-90 s, matrix synthesis has been actively used by many authors to obtain metallic NWs [5,6]. The main questions here are the choice of the template matrix, the metal deposited in its pores and the method of deposition.

Note that other materials can also be used in matrix synthesis. One of the competitors of TMs is porous aluminum oxide (PAO) [8]. Among TMs there are different types of matrixes. Polycarbonate (PC) track matrices are common in Europe. In Russia, membranes made of Polyethylene terephthalate (PET, terephthalic acid ester) are used more often. The use of each of the listed types of membranes has its own specifics and differs greatly for different tasks.

Metals are the main material loaded into the pores of TM [9]. In this case, to obtain NWs, a galvanic (electrochemical) method of metal electrodeposition is used. At the first stage, a metal layer is deposited on one side of the matrix to create electrical conductivity. Such a metallized matrix is placed in a galvanic cell with an electrolyte containing ions of the desired metal (where it works as a cathode). When a current is passed through the cell, the metal is deposited into the pores of the TM. The resulting metal replicas (copy) of pore channels are NWs.

The obtained NWs are in a growth polymer matrix, one side of which is coated with a metal base layer. During subsequent operations, the polymer matrix can be removed, thus forming an array of free-standing NWs, fixed on a common base. In another approach, the base layer can be removed. This forms the so-called metal–polymer composite of many isolated NWs in a polymer matrix. Another option can be the removal of both bases and matrices—in this case, separate NWs are formed.

In such NWs, some of their geometric parameters are determined by the initial growth matrix. Thus, the number of NWs (per unit volume) corresponds to the pore density in the matrix. The diameter of the NWs and their shape are determined by the diameter and shape of the pores in the matrix. The slope of the pores in the matrix determines the geometric orientation of NWs.

Other parameters—the composition of the NWs and the type of NWs—are determined by the composition of the electrolyte and the synthesis parameters. Depending on these conditions, it is possible to obtain one-component NWs (from one metal) or multicomponent NWs.

The simplest type of object, are one-component homogeneous NWs consisting of one metal. The work on matrix synthesis began with this type of NWs. For these objects, changes in growth conditions primarily affect the structure. It has been found that uniform filling of pores is also important, with papers devoted to control of this process in PAO matrices, e.g., [10].

More complex types of NWs are known as multicomponent, consisting of two or more metals. Here two main cases can be distinguished: Homogeneous NWs with a uniform distribution of two (or more) components in the length of the NW, the so-called alloys (alloyed NWs). In this case, growth is carried out in electrolytes containing ions of both (or more) metals, at voltages exceeding the equilibrium voltages for all metals. The second type of structure is heterogeneous: they consist of periodic layers of different elemental composition. This type of structure are called layered NWs. Such NWs can be obtained in two ways, by periodically changing the electrolyte (or transferring the sample to another cell, “bath”), the so-called “two-bath method”. It is also possible to carry out the process in one electrolyte, in one cell, the so-called “one-bath method”. In this case, the electrolyte contains ions of all metals, and a periodic change in the composition of the deposited material is achieved by changing the growth potential. At a lower potential, only one element is deposited—the first—with a lower equilibrium deposition potential (more noble metal). At higher potential, both metals are deposited (but the concentration of the first one could be reduced).

The unique magnetic properties of the samples were detected in the first papers which were devoted to fabrication of Fe-group elements NWs [11]. At first, the most investigated samples were homogenous one-metal NWs [12]. A modern review of magnetic NWs is given in [13].

Alloyed NWs FeCo and FeNi are of great interest. In NWs from alloys, due to a change in composition and/or structure, it is possible to obtain both magnetically hard and magnetically soft, hard materials. The first one can be used to create micro-magnets and magnetic memory elements, while the second can be used for production of magnetic circuits and protective screens. In Ref. [14] FeCo NWs were studied: the saturation magnetization was found to be very high-about 22–24 kG. This makes these wires a potential candidate for deposition of the writer poles in hard disk drives. It was also supposed that the deposition process can be qualitatively divided into two stages. During the initial stage the current decrease with time because of deposition is controlled by planar diffusion due to ion depletion within the pores. Then the diffusion length is decreased (as the pores are filled) which leads to current increase.

In Ref. [15] FeNi and FeCo alloyed NWs (with approximately equiatomic composition) have been electrodeposited in PAO membrane with pores diameter 70 nm. It was shown that the NWs’ composition does not correspond to the content of metals in the electrolyte. The dependence of composition on deposition potential was found. The anomalous co-deposition was also detected, this effect is known as preferential deposition of less-noble metal, in respect to its counterpart. In this case it was preferential deposition of Fe, which was higher in FeNi NWs than in FeCo NWs. The authors also found that this effect is different for the deposition of alloys on the planar electrode and into the pores of PAO matrix. The difference could be explained by the fact that the convection within the pores is more difficult than at the planar electrode, consequently, transport of the less-concentrated ions is more affected. Authors [16] also noted enhancement of the anomalous codeposition in the FeNi alloy when deposited in the nanopores. They explained such behavior by the role of hydrogen bubbling inside the pores, which finally leads to controlling of deposition by mass transport.

In Refs. [17,18] it is shown that by reducing the diameter of FeNi NWs (from 200 to 30 nm), the coercive force can be increased more than 10-fold. In this case, the NWs with the composition of a typical soft magnetic material—permalloy—turn into a hard magnetic material. The coercive force of FeCo changes greatly with a change in composition, and this dependence is not the same as that of a bulk material.

Fe-Co and Fe-Ni NWs were obtained in the pores of PAO in [19]. It was shown that by changing the deposition regimes both types of NWs (layered and alloyed) could be electrodeposited using approximately the same growth solutions. Authors found that cobalt structure in layered NWs depends on the thickness of Co layers: it was FCC for thin and HCP for thick layers. The effect of anomalous co-deposition of Fe in these cases was also detected.

Layered NWs are of great interest too. The effect of dependence of resistance of multilayered NW on an external magnetic field (the so-called giant magneto-resistance, GMR) was described in [20]. The magnetic properties of different types of NW were first reviewed in [21]. Later the possibilities of tuning of NWs’ magnetic properties were supposed in [22]. The application of NWs as sensors for different purposes was proposed in [23].

In Refs. [24,25] NWs with alternating magnetic layers (iron or cobalt) and non-magnetic layers (copper) were studied. It was shown that the nature of the interaction and the magnetic properties depended on the thickness of the nonmagnetic interlayer—the spacer. The alternation of metals with different magnetic properties made it possible to change the spin polarization for passing current and to use these NWs as elements of spintronics. In [26], it was proposed to record bits of information “along the length” of single layered NW in order to increase recording density.

In layered NWs the lengths of the layers and the nature of their alternation can be changed. Such NWs are of interest as elements of electronics, spintronics, sensors, and valves [27]. In [24] it was shown that their magnetic properties also depend on the ratio of the thicknesses of the magnetic and non-magnetic layers. By changing this ratio, the orientation of the easy magnetization axis can be changed. Additionally, in multilayered NWs all segments must have the same compositional and geometrical parameters. In some cases, the NWs with very thin layers—less than the spin relaxation length for electrons—are needed. Moreover, some magnetic transport effect could be obtained only in NWs with definite borders between layers (the so-called interfaces), and these borders must be thin and flat.

To summarize these reviewed papers, it could be concluded that there are many published data, but they are often contradictory. The main point here is the features of electrodeposition into narrow pore channels. Systematic data on the relationship between structure and growth conditions is needed. In the case of magnetic metals, the composition and the structure of the obtained nanowires will obviously determine their unique magnetic properties. Therefore, the aim of this work is to compare different ways of fabricating NWs and to find new approaches to managing their structure.

## 2. Materials and Methods

Track-etched membranes (fabricated in JINR, Dubna) with cylindrical through-pores were used as templates. The membrane parameters were as follows: film thickness was about 10 μm, the pore diameter in different membranes was 70–500 nm, and the pore density was 10^8^–10^9^ pores per.cm^2^. To carry out electrodeposition, one of membrane sides was coated with a 2–10-μm-thick continuous conducting copper layer. The last one was deposited by a two-step technique: thermal evaporation of copper (formation of thin Cu layer) was followed by galvanic deposition of copper (formation of thick layer).

Electrodeposition of metals into the pores was carried out in a special home-made galvanic cell. This cell provided vertical fixation of a prepared matrix with a conductive layer and its electrical connection to conducting wires. Different types of anodes were used (see below). The anodes were immersed vertically in a galvanic cell with electrolyte. Deposition was performed in the two-electrode geometry on a sample area of 2.5 cm^2^. An Elins P-2X potentiostat–galvanostat was used as a current source. The process was carried out in the galvanostatic mode at the constant potentials of different values (see below), at room temperature. During the electrodeposition, the time dependence of current was recorded, due to which the process could be controlled.

The compositions of electrolytes differed depending on the experiment. In general, as a metal-ion source, the corresponding metal salts were used, usually sulfate and chloride salts. These salts are highly soluble, and their concentration could be varied in wide range.

For example: the solution for producing NW from FeNi alloys contained NiSO_4_ ∙ 7H_2_O (16 g/L), NiCl_2_ ∙ 6H_2_O (40 g/L), and FeSO_4_ ∙ 7H_2_O (from 4 to 32 g/L). Electrolytes for formation of FeCo NWs contained CoSO_4_ · 7H_2_O (16 g/L) and CoCl_2_ · 6H_2_O (40 g/L). The concentration of FeSO_4_ · 7H_2_O was varied from 4 to 72 g/L, which correspond to a change in the iron-to-nickel ionic ratio from 6% to 53%. To increase the relative iron ion concentration more significantly, the cobalt salt concentrations were reduced as follows: 12, 8, 4, and 2 g/L for CoSO_4_ ·7H_2_O and 32, 24, 16, 8, and 4 g/L for CoCl_2_·6H_2_O. This technique made it possible to change the relative iron ion concentration from 59% to 91%.

Some additives were used in all types of electrolytes: Lauryl sylfate, for wetting and Boric acid, for pH stabilization (25 g/L). In iron-containing electrolyte ascorbic acid (1 g/L) was used in order to prevent Fe^2+^ ions oxidating to Fe^3+.^

The typical regimes for electrodeposition of alloyed NWs were: voltages changing from 1 to 2 V, depending on the experiment; Fe was used as anode; and the duration of pulses was selected from the chronoampere dependences.

For obtaining of layered NWs: voltage for Cu deposition −0, 5 V, and 1.5–1.8 V for deposition of Ni (or Co). In the case of time-control, the deposition time was calculated from the thickness of the layer to be deposited. In the case of charge control, for example, a charge flow of 271 mC is required for Cu layers and 287 mC for magnetic layers.

For further investigations, the NWs were removed from the polymer. In order to release NWs from the host membrane, the last one was dissolved in alkali solution (5N NaOH, t = 50 °C, 2 h). Chemical removal of the polymer was followed by washing of the NW sample in distilled water.

Scanning electron microscopy (SEM) with energy-dispersive X-ray analysis (EDXA) was performed on a JEOL JSM 6000 plus microscope (JEOL, Japan) in the secondary-electron mode at an accelerating voltage of 15 kV. X-Ray diffraction (XRD) analysis was carried out on a RIGAKU MiniFlex 600 diffractometer (Rigaku, Japan) using CuKα radiation, at 2θ angles in the range 40°–80° with a scan step of 0.01° and a rate of 1 deg/min.

The structural studies of individual NWs were performed by transmission electron microscopy (TEM), high-resolution TEM (HRTEM), scanning TEM (STEM), with a high-angle annular dark field (HAADF) detector and electron diffraction using an Osiris microscope (Termo Fisher Scientific, USA) at an accelerating voltage of 200 kV. The elemental analysis and distribution maps of chemical elements were obtained using a special EDXA system SuperX equipped with four Si detectors. The images and electron diffraction patterns obtained in the electron microscope were processed and analyzed using the DigitalMicrograph, Esprit, TIA, and JEMS software.

## 3. Results and Discussions

### 3.1. One-Component NWs

At the first stage of the experiment the matrix (track membrane) was always tested. An SEM image of membrane surface is given in Figure 1a. The control deposition process was carried out until over-growth of metal “out of the pore channels” which leads to formation of so-called “caps”. SEM-image of the membrane surface with metal caps is given in Figure 1b.

The NWs’ topology, diameter, and length could be estimated after the release of wires from the matrix: see Figure 1c and description below.

Cobalt NWs. Cobalt structures are of interest as magnetically hard materials with high saturation magnetization and high Curie temperatures. A feature of cobalt is also the fact that the structure of the resulting material can change depending on the synthesis conditions. It is assumed that in one-dimensional nanomaterials, e.g., NWs, it is possible to vary the properties and their anisotropy by changing their size.

Growth of cobalt NWs: This work continues the study of the effect of pore diameter and growth voltage on the dynamics of NW growth, which was started in [28]. The features of the deposition of cobalt into a matrix with pores with a diameter of 100 to 500 nm with the growth voltage varied from 500 to 700 mV have been studied (using the two-electrode growth scheme). The obtained graphs are presented in Figure 2.

Obtained dependences of current versus time (in potentiostatic mode) showed that deposition at the stage of filling the pore channel is nonlinear. At the beginning of the process, its speed decreases but after partial filling of the pores, it begins to increase. The first effect can be explained by the depletion of the electrolyte during the growth of NWs, which arises due to the slow delivery of metal ions to the growing NWs under conditions of narrow and long pore channels. The second effect can be associated with a decrease in the electrical resistance of the pore channel as it is filled with metal and a corresponding decrease in the length of the electrolyte layer. It can be assumed that the final speed is determined by the competition between the two processes. In this case, at the initial stage, the first effect predominates, and in the final stage, the second effect is the main one. It is obvious that changing of the ion mobility in a narrow pore channel also plays a certain role. A strict confirmation of this explanation is that the effect is significantly increased in narrow pore channels. The effect also increases with increasing growth speed. (That is, in narrow pores and with an increase in the growth rate, the depletion of the electrolyte occurs faster).

The resulting arrays were examined by SEM—an example is shown in Figure 3a.

It is noticeable that the real diameter of nanowires is larger than the pore diameter of the growing matrices. For example, NWs grown in a matrix with pores 100 nm in diameter will have a diameter of approximately 110 nm. This difference could be explained by the surface oxidation of the NWs. Another explanation is that the growing metal is stretching the polymer matrix.

In Ref. [29] the effect of the surface density of pores on the features of their filling was investigated for cobalt NWs. It is known that the nature of the process is influenced by the near-surface diffusion layer. In the case of matrix synthesis, the diffusion layer includes a region within a pore and an electrolyte area adjacent to the “mouth” of the pore. The value of the first component changes with the growth of NWs from 10 μm at the beginning of growth (practically the entire pore length) to zero (at the end of growth, when the NW has filled the entire pore channel). The value of the second component depends on whether the diffusion areas of adjacent pores overlap at the matrix surface. According to the theory proposed in [29], two very different cases are possible. With a relatively low pore density, these areas of adjacent pores do not overlap and can be represented as hemispheres with a radius slightly exceeding the pore radius. In this case, a relatively small value is added to the diffusion layer inside the pore, equal to the radius of such a hemisphere. In the second case, with a high pore density and, accordingly, a small interpore distance, neighboring hemispheres overlap. This leads to the diffusion layer becoming continuous and covering the entire surface. Therefore, its effective thickness increases many times. The character of the filling of the pores in these two described cases will be different. In the first case, during the pore filling, the relative change in the length of the diffusion layer will be significant. This will cause a change in the conditions of electrodeposition as the NWs grow. In the second case, the total thickness of the diffusion layer will be determined mainly by the area outside the pores (over the membrane surface). In this case, as the NWs grow, the relative decrease in the thickness of the diffusion layer will be much smaller. It can be assumed that as a result of this, in the second case, the deposition conditions during the growth of nanowires will change significantly less. The nature of the deposition will become more uniform, which was observed in [29]. At the same time, a difference in the deposition rate was revealed; in the second case, an increase in the effective thickness of the diffusion layer led to a slowdown in the growth rate.

The growth and structure of cobalt NWs in various electrolytes was studied in [18]. In this work it was shown that, depending on the pH of the growth solution, either cubic (pH less than 4) or hexagonal modification (pH greater than 5) can be obtained. The results of X-ray analysis are shown in Figure 3b. The difference between two spectra could be explained by dominance of the cubic FCC phase for samples grown at low pH = 3 and by dominance of the hexagonal HCP phase obtained in samples obtained above pH = 5 (Note that in both cases admixture of second phase also present).

The effect of the influence of the electrolyte acidity was observed earlier for the growth of bulk samples; however, a change in the type of structure was observed at other pH values. The data presented in some papers ([30], for example]) show that for Co NWs, the electrodeposition voltage can also affect the structure and phase composition of wires and, therefore, their magnetic and conductive properties.

### 3.2. Multicomponent NWs

Synthesis of two-component (or many-component) nanowires has all the above-described features. In addition, new features associated with different equilibrium deposition potentials, diffusion differences, and differences in the character of deposition appear.

#### 3.2.1. NWs from Alloys

Two types of structures from alloys were synthesized: iron-nickel NWs and iron-cobalt NWs, both with different ratios of metal concentrations. The concentration ratio was changed by changing the composition of the growth electrolyte (see Section 2). The dependence of current on time was plotted. These graphs allowed control of the process of deposition. The obtained graphs are given in Figure 4a.

It should be noted that this graph looks almost the same as the graph for deposition of pure metal. In our earlier works [31,32] a relationship was revealed between the ratio of the concentrations of metals in the growth electrolyte and in the obtained NWs. Thus, it has been shown that in iron-cobalt NWs, the ratio of metal concentrations approximately corresponds to their ratio in the electrolyte (the difference is no more than 3−7%). Note that this ratio is hardly affected by other growth parameters, for example, the growth voltage (within the voltage range from 1 to 2 V). Microscopic images of NW (TEM) were obtained. A TEM-image of FeNi NWs is presented in Figure 5: HAADF STEM image and corresponding EDX-maps of element distribution.

A TEM image of FeCo NWs is presented in Figure 6: HAADF STEM image and corresponding EDX-maps of elements distribution. The electron diffraction images are given in Figure 7.

It could be concluded that the composition of FeCo NWs does not practically change along the length of the NWs. The formation of iron oxides at the NW’s surface was detected.

At the same time, the ratio of the concentrations of metals in iron-nickel NWs differs markedly from the ratio in the growth electrolyte. In all cases, the iron content in the NWs is significantly higher than in the electrolyte. The magnitude of this effect depends on several factors. Thus, the difference increases with an increase in the initial concentration of iron. In addition, the iron content in the NW increases with a decrease in the growth voltage. Finally, it was found that the concentration ratio changes along the length of the NW: at the “top” of the NW (at the end of the growth process), the iron content increases. The concentration difference along the length can be up to 20%. This was concluded based on the data of local elemental analysis (in SEM). It should be noted that the above estimations were given for an average amount of iron averaged over the length of the NW.

This phenomenon, which was previously discovered during the growth of bulk alloys, as well as in some cases during the growth of NWs in PAO (see, for example, [15]), is often called anomalous co-deposition of iron. The origin of anomalous co-deposition in the iron group metals is under debate and two mechanisms have been proposed: changing of pH just near the surface and the formation of metal hydroxides. In our mind an additional reason for this may be the difference in the mobility of nickel and iron ions—the latter have higher mobility. At the same time, the mobility of cobalt and iron ions are approximately the same, which explains the approximate correlation between the compositions of FeCo NWs (determined by EDX) and calculated composition of the grown electrolyte.

In our work, it has been shown that the effect of increase of the rate of deposition of iron in the pores of TM is higher than that of electrodeposition on a flat surface. This can be explained by the difference between the mobility of iron and nickel ions in narrow pore channels.

#### 3.2.2. Layered NWs

In the present work, studies on the processes of obtaining layered NWs, started in [33], are continued. NWs were synthesized with alternating layers of magnetic metal (cobalt or nickel) and non-magnetic copper: Co/Cu or Ni/Cu. Layered NWs were obtained by a single-bath method—the electrolyte contained ions of both metals, voltage pulses of a certain magnitude and duration led to the predominant growth of the magnetic metal or copper.

Solutions of sulfate salts of the corresponding ions were used as electrolytes; it should be noted that the concentration of copper ions was 20 times less than that of the magnetic metal. This is necessary to reduce the concentration of copper in the deposited magnetic layer. For the deposition of pure copper layers, a potential of 0.5 V was applied. This made it possible to deposit copper at a critical current, preventing the electrochemical deposition of nickel or cobalt. For the deposition of the magnetic layer, a potential of more than 1.5 V was applied. This potential (together with the metals concentration ratio) ensured deposition with a minimum amount of copper impurities. Dependence of current on time for this deposition process is given in Figure 4b. Figure 8 shows SEM images of the obtained layered NWs.

Some results of our TEM investigation of Co/Cu and Ni/Cu NWs are given in [34] Here some new results are presented. All types of observed NWs (alloyed and layered) were polycrystalline. No amorphous phase was observed. This was evidenced by TEM data and X-ray diffraction patterns as well. In the case of XRD patterns, the peaks were broadened due to the small size of the crystallites. An example of HAADF STEM imaging and corresponding EDX mapping for Co/Cu NWs is shown in Figure 9.

Another type of NWs, Ni/Cu, are studied in more detail. The TEM microscopic image of Ni/Cu NWs with bright-field and dark-field images are presented in Figure 10.

The grain structure of synthesized nanowires can clearly be seen in TEM micrographs of their fragments (Figure 10). The contrast in bright-field TEM image indicates that the nanowires are polycrystalline. To study the morphology of grains and identify the phase composition of nanowires, we performed a dark-field TEM and diffraction analysis of their individual fragments (Figure 10b–e). Crystallites of different sizes were observed: small (5–20 nm, Figure 10b,c) and relatively large (to 100 nm, Figure 10d,e). An analysis of interplanar spacings in diffraction patterns revealed that they correspond to several phases. Large bright reflections are mainly due to metals, Ni (Fm3m) and Cu (Fm3m), while diffuse rings and small reflections are generally related to copper oxides of different stoichiometry: Cu_2_O and CuO.

The main goal at this stage was to reduce the thickness of the layers. At the same time, the tasks of controlling the composition of individual layers, the thickness (length) of layers of the same type, as well as the repeatability of these parameters along the length of the NWs were solved. The shape of the boundary between the layers and the possibility of its control were also studied. Initially, in order to reduce the thickness of the layers, the deposition time of each layer was reduced from 150 s and 20 s to 15 s and 2 s (for Cu layer and Ni layer, respectively).

Most of the experimental results obtained in this work are illustrated by TEM images of Ni/Cu NWs. (It should be noted that the same experiments were carried out for Co/Ni NWs too and similar dependences were obtained). TEM data for the obtained Ni/Cu NWs are given in Figure 11.

The images in Figure 11 show that the thickness of the layers changes along the length of the NWs. Elemental analysis showed that with a decrease in the thickness of the layers, a redistribution of elements occurs between adjacent layers—each layer contains a noticeable amount of metal from the adjacent layer. The boundary (interface) between the layers was not flat: it’s surface became uneven, with one layer ‘flowing’ into the region of another. To overcome these effects, some techniques used in the growth of multilayer bulk materials were applied.

The next approach was the control of the passed charge. Note that controlling the layer growth over the time of the corresponding pulse duration, which was used at the initial stages, is the simplest way to control the thickness. However, when carrying out the process in a potentiostatic mode, a galvanic cell with electrolyte changes its resistance. This approach led to the amount of deposited metal changing from layer to layer. In this regard, the method of monitoring the passed charge was applied—the pulse duration was determined by the amount of electricity passed. This parameter, in accordance with Faraday’s law, gives control of the amount of deposited material. TEM images of the obtained NWs are shown in Figure 12.

The image in Figure 12 demonstrates that the thickness of the layers is approximately the same along the entire length of the NWs.

The effect of pauses between the pulses of deposition of individual layers was also investigated. It was assumed that a short-term complete switching of the voltage between successive pulses of different voltages will make it possible to compensate the depletion of the electrolyte near the growing surface by keeping up the concentration. Therefore, pauses with a duration of 5 s were used. Note that the experiments tested the effect of pauses both after the growth of each layer and only after the deposition of the magnetic layer. The microscopic image of the obtained nanowires is shown in Figure 13.

From the images in Figure 13, it can be concluded that the use of pauses led to an improvement in the shape of interlayer boundaries (interfaces). At the same time, there was an uncontrolled change in the thickness of the layers. It can be assumed that this also happened because of the dissolving of the deposited magnetic layer. One of the reasons for the observed effects may be the appearance of an open circuit potential difference.

To obtain smoother and flatter boundaries (interfaces) between layers, additives used in the deposition of bulk materials were tested. It is known that Butanediol is used as a smoothing agent in the deposition of nickel. Butanediol concentrates on the surface protrusions and slows down the electrodeposition there. This leads to a relative acceleration of the process in the cavities and thus leads to the smoothing of the deposited layer. TEM images of NW obtained with the addition of Butanediol to electrolyte (1 g/L) are shown in Figure 14.

Figure 14 demonstrates that the interlayer boundary becomes quite even, the nickel layers have a compact appearance. On the other hand, the copper layers become loose and have a porous structure. Perhaps this is due to the fact that at a low concentration of copper, Butanediol has a strong and uncontrolled effect on its deposition–so it’s deposition has local character.

During the electrodeposition of copper, a brightening (“shining”) organic additive (Russian brand ZKN-74) is sometimes used. In massive samples, its use leads to grain refinement, the formation of a fine-crystalline structure and, accordingly, to the leveling of the surface. In this work, ZKN was added to the standard composition at a concentration of 1 g/L. Examples of the obtained NW TEM images are shown in Figure 15.

The presented images show that the layers (both copper and nickel) in the NWs have the correct flat shape. At the same time, the thickness of the layers obtained in some cases is higher than calculated. This can be explained by a non-uniform filling of different pores: if some of the pores are not filled (due, for example, to bubble-blocking), the quantity of metal in each layer of the other pores should increase. In general, the use of this additive gives a positive result. However, in this case, the calculation regime must be corrected.

Another method for obtaining NWs with thin layers is to reduce the concentration of ions of the deposited metals in the electrolyte. It is known that in order to decrease the layer thickness, the time of the deposition pulse for a single layer must decrease. Therefore, to obtain layers of several nanometers in the usual mode, it is required to use pulses with a duration of the order of a second or less. However, this time is comparable to the time of the equilibrium state. The use of dilute electrolytes has been suggested. The percentage of both ions was decreased by eight times. Note that with a decrease in the concentration of metal salts, the concentration of additives did not change. This method made it possible to increase significantly the time for the deposition of individual layers so that the process was better controlled. An example of the NWs obtained is shown in Figure 16.

An analysis of the results obtained showed that the use of solutions with a lower ion concentration made it possible to reduce the layer thickness to 13 nm. However, the boundaries of the layers were uneven. Moreover, the total deposition time in this case increased significantly.

In some experiments, a decrease in the duration of growth pulses led not to a decrease in the layer thickness, but to the formation of a ‘core–shell’ structure. For example, during the deposition of nickel-copper NWs with a decrease in the deposition time, a structure of copper rods coated with nickel was formed. One metal was ‘covered’ by the other. Instead of obtaining of layered NWs with the calculated layer thickness of less than 10 nm, ‘core–shell’ NWs were obtained. An example of the resulting structures is shown in Figure 17.

The formation of such structures can be attributed to the interaction of metal ions with the pore walls—the adhesion of the nickel ions to the pore wall in the polymer is higher than that of the copper ions. Another explanation is the features of the transient processes that occur when the potential changes. When switching the potential from a larger to a smaller one, the device registered the occurrence of a reverse current, which indicated a partial dissolution of the just-deposited metal. It can be assumed that mainly the central area of the grown nickel layer was dissolved. In the next cycle, this area was filled with copper. One of these effects or their combination could be the reason for the formation of structures of the ‘core–shell’ type.

Another approach is to introduce a third electrode into the galvanic cell circuit. The reference electrode made it possible to control the potential directly near the surface of the template membrane. In this case, the potential itself is more accurately set; in the deposition process, the electromotive power of the broken circuit is taken into account at each potential switching. Deposition according to the three-electrode scheme made it possible to obtain thinner layers. Figure 18 shows TEM images of NW with layers of decreasing thickness. The thickness was set by the charge, and its calculated value was halved in each subsequent layer.

The above images proved the possibility of precise control of the layer thicknesses. The minimal thickness of the obtained layers was 7–10 nm.

## 4. Conclusions

In this work, we investigated the features of the process of obtaining one-component and two-component nanowires by electrodeposition into the pores of track membranes. The work was carried out for metals of the iron group, the nanoscale structures of which can have unique magnetic properties.

Using one-component NWs as an example, the origins for the nonlinearity of the galvanic process are revealed: the competition between the processes of depletion of the electrolyte due to the slowing down of the supply of ions and the change in the thickness of the electrolyte layer in the pores. It was shown that, at a high pore density, the areas of diffusion supply of ions from neighboring pores overlap, which leads to the appearance of a rather thick continuous diffusion layer above the membrane surface. The presence of this thick layer leads to the stabilization of the process of electrodeposition. For cobalt NWs, it was shown that an increase in the acidity of the electrolyte leads to a transition from a cubic to a hexagonal structure.

For two-component structures, the processes of obtaining nanowires from FeNi and FeCo alloys have been studied in detail. For nickel compositions, the effect of predominant deposition of iron (anomalous co-deposition of Fe) was found. The latter is significantly enhanced in pores of small diameter and can be associated with the lower mobility of nickel ions. Methods for obtaining layered nanowires with thin layers were considered. A number of techniques were investigated, including electrolyte dilution and the use of additives. It was shown that the most effective methods are the use of a three-electrode cell and the use of a control of the passed charge mode. At the same time, it has been shown that a number of techniques that are successfully used for the growth of bulk materials (e.g., using a pause between current pulses) are inefficient when carrying out the process in narrow pore channels.

The obtained results allow one to vary the geometry of NWs (obtained in the pores of track membrane), their composition and structure and, consequently, their magnetic properties. The possibily of obtaining narrow NWs and/or NWs with precise control of composition enables NWs with high coercive force (Hc). For alloyed NWs, the effect of anomalous codeposition of iron and the factors leading to it were investigated. This effect (which is higher for FeNi NWs) should be taken into consideration when alloyed NWs are produced. Layered NWs with rather thin layers have high magneto-resistance properties. Controllable changing of the layer thickness allows control of the magnetic properties.

All results make it possible to have better control over the structure of the obtained nanowires, which determines their magnetic properties.

## Figures and Tables

**Figure 1 membranes-12-00195-f001:**
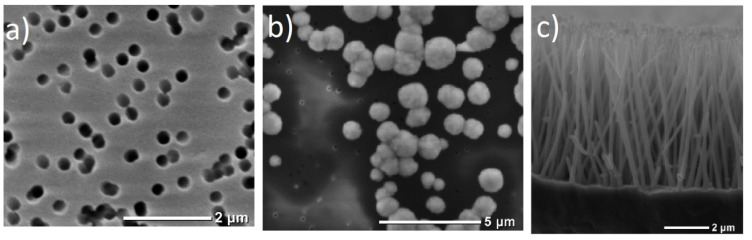
SEM-images: (**a**) top-view of the membrane surface; (**b**) metal “caps” at the surface of TM; (**c**) NWs after removal of the polymer matrix.

**Figure 2 membranes-12-00195-f002:**
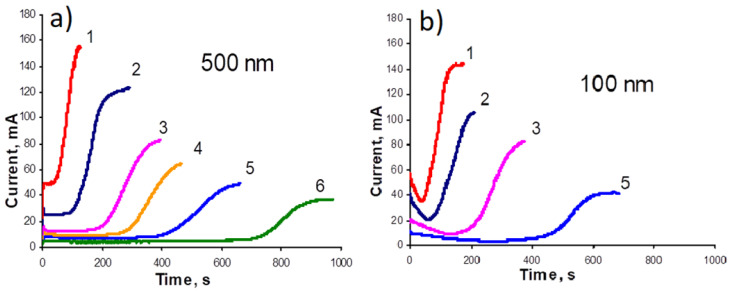
Potentiostatic curves for deposition of pure Co into the pores: (**a**) pore diameter 500 nm, (**b**) pore diameter 100 nm (deposition voltage: 1–680 mV, 2–630 mV, 3–580 mV, 4–555 mV, 5–530 mV, 6–505 mV).

**Figure 3 membranes-12-00195-f003:**
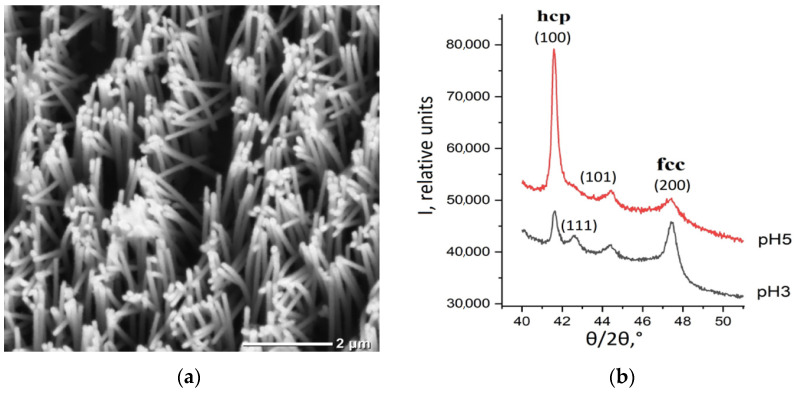
(**a**) SEM image of arrays of Co NWs; (**b**) XRD data: diffractograms of the Co NWs grown in electrolytes with pH = 3 and pH = 5.

**Figure 4 membranes-12-00195-f004:**
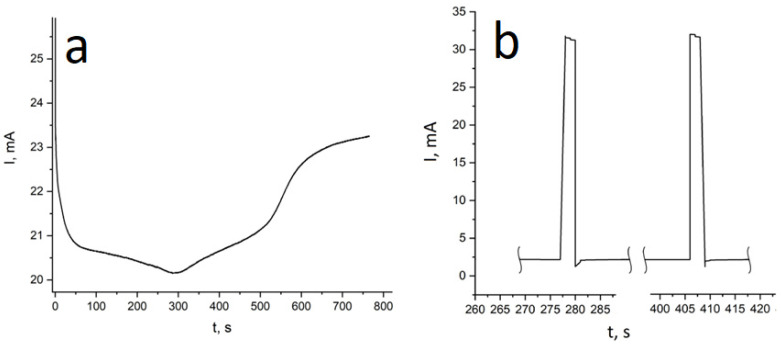
Deposition curves: (**a**) Potentiostatic curves for deposition of FeCo alloyed NWs; (**b**) Dependence of current on time for pulsed deposition (obtaining Ni/Cu layered NWs).

**Figure 5 membranes-12-00195-f005:**
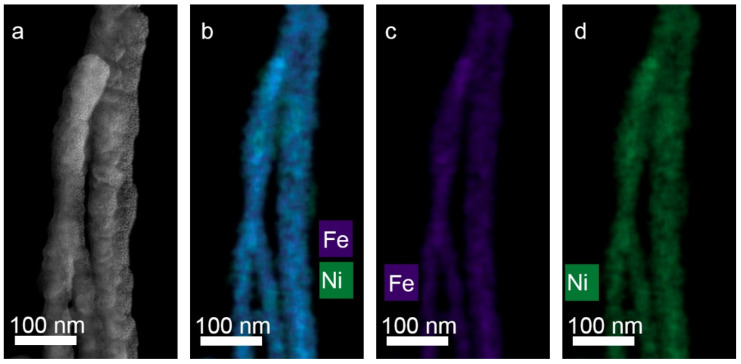
HAADF STEM image (**a**) and corresponding EDX mapping (**b**–**d**) for alloyed FeNi nanowires.

**Figure 6 membranes-12-00195-f006:**
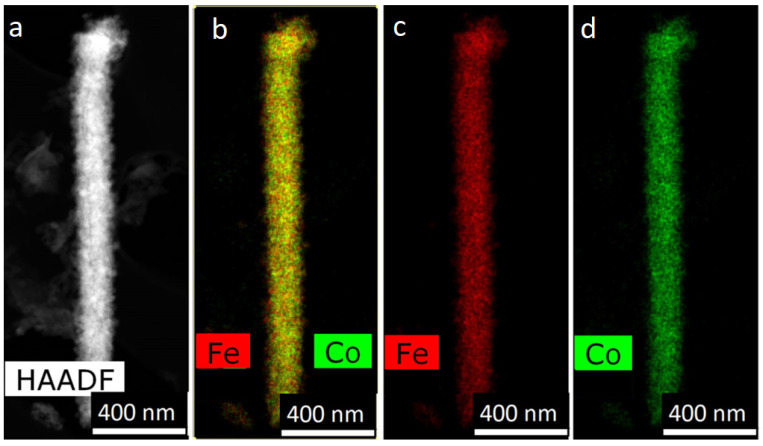
HAADF STEM image (**a**) and corresponding EDX mapping (**b**–**d**) for alloyed FeCo nanowires.

**Figure 7 membranes-12-00195-f007:**
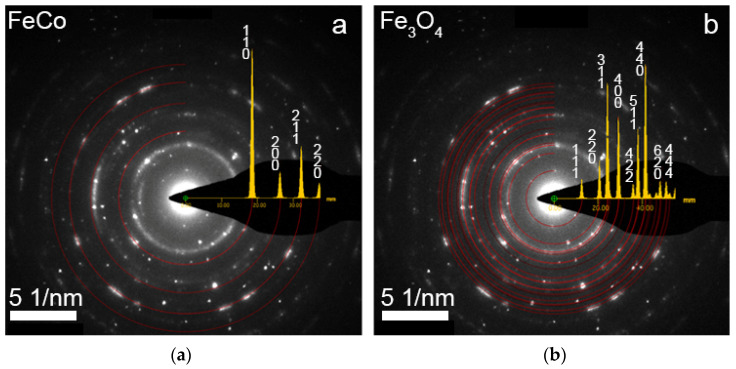
Selected area electron diffraction (SAED) pattern of FeCo NW and model ring electron diffraction patterns for FeCo phase (**a**), Fe_3_O_4_ phase (**b**).

**Figure 8 membranes-12-00195-f008:**
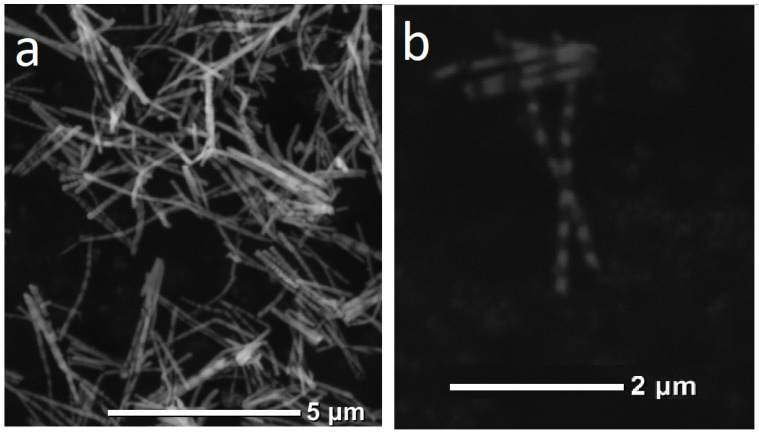
SEM images of layered Ni/Cu (**a**) and Co/Cu (**b**) NWs.

**Figure 9 membranes-12-00195-f009:**
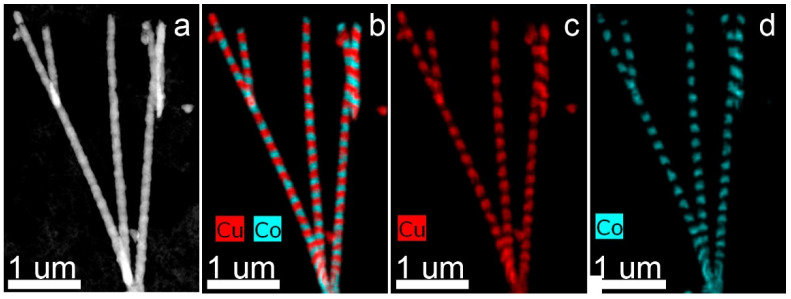
HAADF STEM image (**a**) and corresponding EDX mapping (**b**–**d**) of layered Co/Cu NWs.

**Figure 10 membranes-12-00195-f010:**
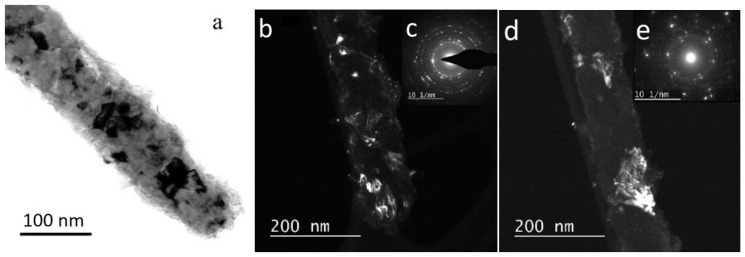
TEM: Structure of individual Ni/Cu NWs: (**a**) bright-field image of grains, (**c**,**e**) insets with electron diffraction patterns; (**b**,**c**) dark-field images of small crystals and the corresponding electron diffraction pattern, and (**d**,**e**) images of an individual large grain and the corresponding electron diffraction pattern.

**Figure 11 membranes-12-00195-f011:**
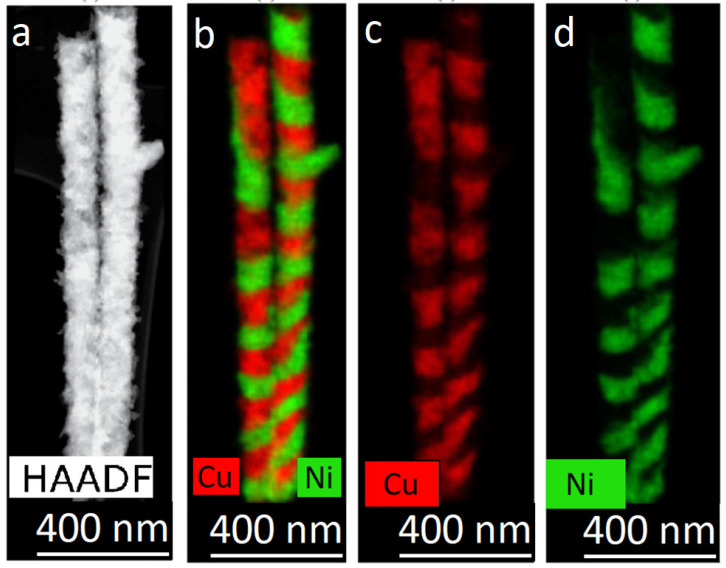
HAADF STEM-image (**a**) and corresponding EDX- mapping (**b**–**d**) of layered Ni/Cu NWs, obtained with pulse-time control.

**Figure 12 membranes-12-00195-f012:**
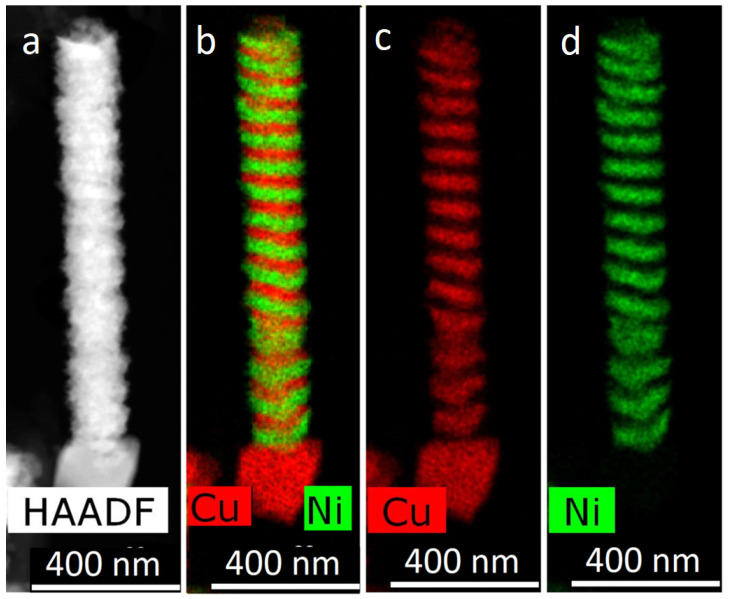
HAADF STEM image (**a**) and corresponding EDX mapping (**b**–**d**) of Ni/Cu NWs, obtained with control of the passed charge.

**Figure 13 membranes-12-00195-f013:**
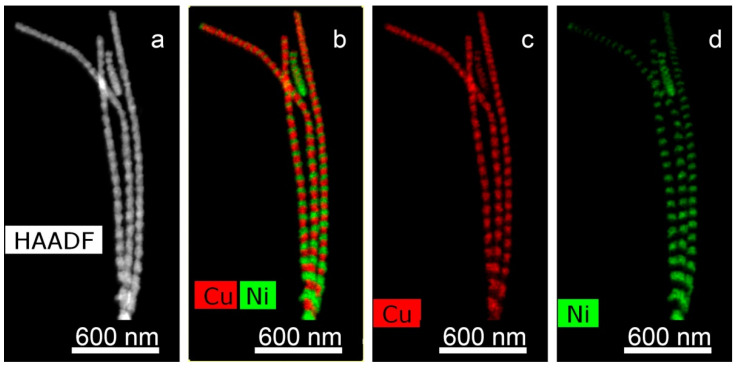
HAADF STEM-image (**a**) and corresponding EDX mapping (**b**–**d**) for Ni/Cu NWs grown with pauses between growth pulses.

**Figure 14 membranes-12-00195-f014:**
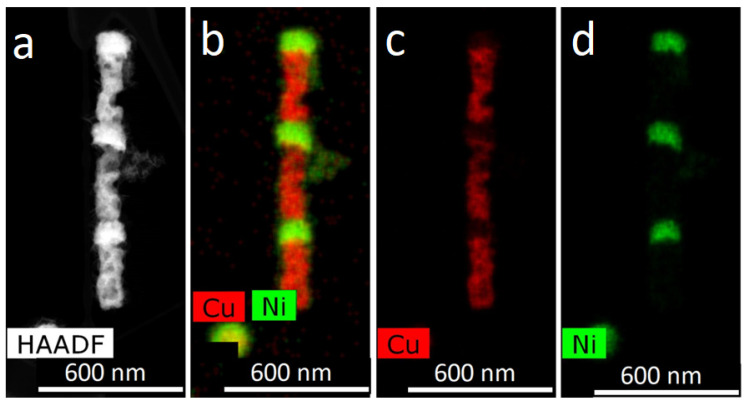
HAADF STEM-image (**a**) and corresponding EDX mapping (**b**–**d**) for Ni/Cu nanowires, grown in an electrolyte with the addition of Butanediol.

**Figure 15 membranes-12-00195-f015:**
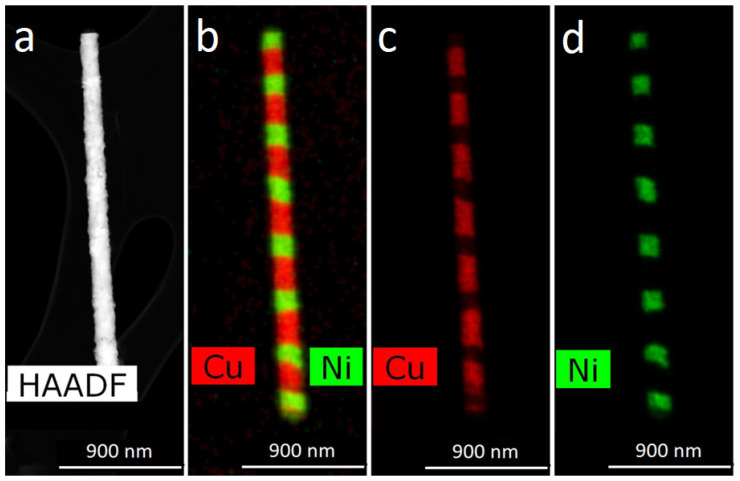
HAADF STEM image (**a**) and corresponding EDX mapping (**b**–**d**) for Ni/Cu nanowires grown in an electrolyte with the use of a brightening additive.

**Figure 16 membranes-12-00195-f016:**
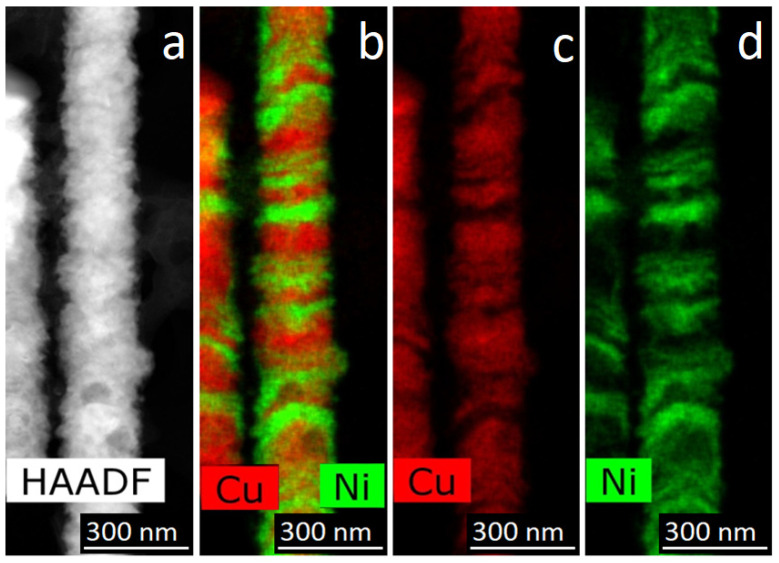
HAADF STEM image (**a**) and corresponding EDX mapping (**b**–**d**) for Ni/Cu NW with thin layers, obtained using dilute electrolytes.

**Figure 17 membranes-12-00195-f017:**
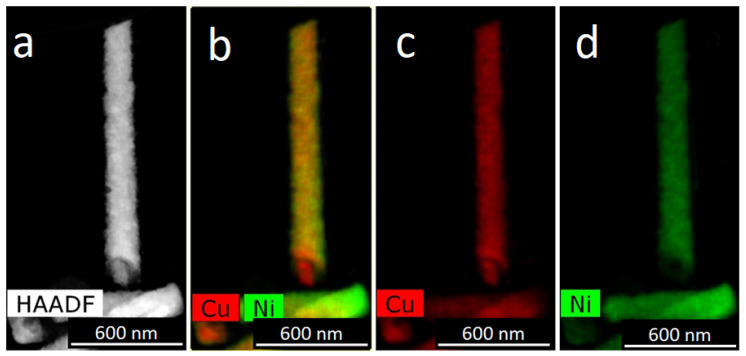
HAADF STEM image (**a**) and corresponding EDX mapping (**b**–**d**) for Ni/Cu nanowires with ‘core–shell’ structure formed during synthesis with short growth pulses. One can see the difference in diameter between Cu and Ni images.

**Figure 18 membranes-12-00195-f018:**
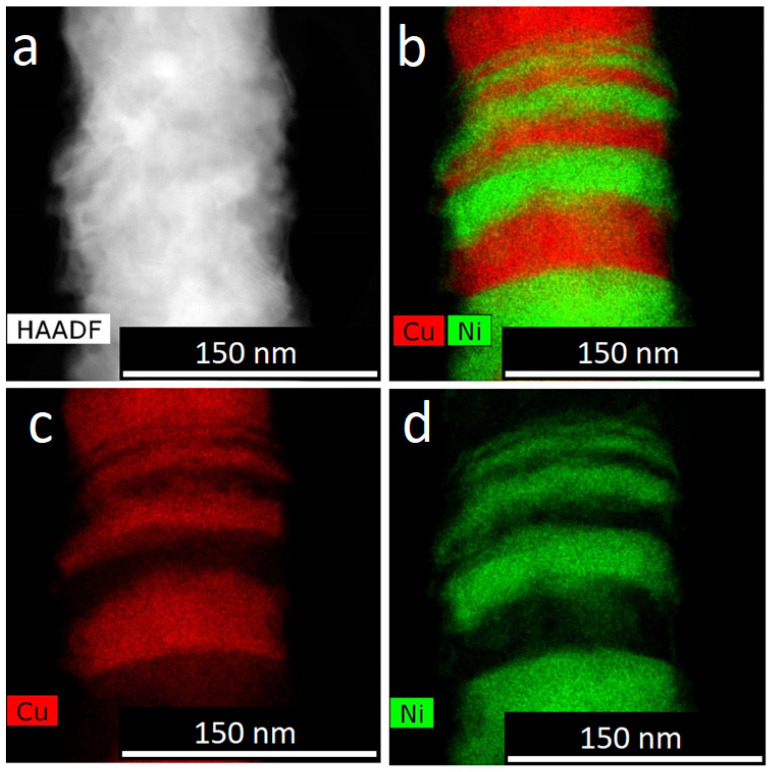
HAADF STEM image (**a**) and corresponding EDX mapping (**b**–**d**) of Ni/Cu NW with a sequential decrease in the layer thickness. Deposition with reference electrode Ag/AgCl.

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
