# Peer review of "Formation of Nanowires of Various Types in the Process of Galvanic Deposition of Iron Group Metals into the Pores of a Track Membrane"

_membranes, 2022, doi:10.3390/membranes12020195_

Round 1
Reviewer 1 Report
Reviewer has the following comments:
1) On page 6, the authors make the following statement: “It is noticeable that the diameter of nanowires with small diameters (100 nm and less) is larger than the pore diameter of the matrices”. This statement is not quite clear, since the pore diameters range from 100 to 500 nm (see page 5) “… into a matrix with pores with a diameter of 100 to 500 nm”.
2) On page 7, the authors make the following statement: “Preliminary data also show that, with the growth of NWs with a hexagonal structure, the electrodeposition conditions can also affect the orientation …”. It is not quite clear what kind of preliminary data are discussed. The authors should either add this data or not discuss this result.
3) Page 8, Figure 3. Images of electronic diffraction (a) and (b) look the same. At the same time, the sections of diffraction are different. The authors should explain in the text the interpretation of these data in more detail. Sections of diffraction should be added as separate figure.
4) On page 8, the authors make the following statement: “And, finally, it was found that the concentration ratio changes along the length of the NW: at the “top” of the NW (at the end of the growth process), the iron content increases. The concentration difference along the length can be up to 20%”. It is not quite clear how this result was obtained. What experimental data confirm this result?
5) On page 8, the authors make the following statement: “… which explains the approximate correlation between the compositions of FeCo NW and the composition of the electrolyte”. What experimental data confirm this correlation?
6) On page 12, the authors make the following statement: “…the thickness of the layers obtained is much higher than the calculated one. This can be explained by a change in the electrodeposition mode.” The authors should explain this in more detail.
7) On page 15, the authors make the following statement: “It is shown that, at a high pore density, the regions of diffusion supply of ions from neighboring pores overlap, which leads to the appearance of a rather thick continuous diffusion layer above the membrane”. What experimental data confirm this result?
8) The text contains a number of typos (page 5 “… from the ratio that …”). There are some inaccuracies in English (page 7, EDX-cards -> EDX-maps, etc.). Some words are written with a capital letter (page 4 “Different types of Anodes”, etc.). There are double points at the end of the proposals (page 7 “with respect to the nanowire axis..” etc.). A number of sentences in the text are taken into brackets. Some abbreviations are not “widened” in the text (page 5 - SEM, TEM). Section 5 "Patents" is empty.
Author Response
Dear Reviewer! Thank you for reviewing of our paper! Your suggestions are very important and made it possible to improve the text.
We have corrected our text according to your recommendations. Please, see the attachment.
Sincerely yours- Dmitry Zagorskiy

Reviewer 2 Report
The work submitted by Zagorskiy describes "Formation of nanowires of various types in the process of galvanic deposition of iron group metals into the pores of track membrane". The manuscript has well-written and well-organized. Also, the connections and correlations between different parts are well established. I think by considering below points, this manuscript can be published in membranes.
- There are a quarter citations to Please consider it. These references would be also useful:
- Electrodeposited Ni-Rich Ni–Pt Mesoporous Nanowires for Selective and Efficient Formic Acid-Assisted Hydrogenation of Levulinic Acid to γ-Valerolactone: https://doi.org/10.1021/acs.langmuir.1c00461
- Electrochemical pore filling strategy for controlled growth of magnetic and metallic nanowire arrays with large area uniformity: https://doi.org/10.1088/0957-4484/27/27/275605
- A SEM top view image of TM with membranes would give better vision of work. Also the lengths and therefore the aspect ratios of NWs are unknown. Please show the whole lengths of nanowires.
- How the authors release the NWs from membrane? Explain the release mechanism.
- Are the membranes filled together? In other words how much is pore filling? It is better to show the top surface of electrodeposited membranes.
- The authors must show the growth mechanism of single, alloy and layered nanowires with the electrodeposition profiles meaning the reduction potential vs time curves.
Author Response
Dear Reviewer!
Thank you for reviewing of our paper! Your
We have corrected our text according to your and other reviwers recommendations. -Please, see the attachment.
Sincerely yours-Dmitry Zagorskiy

Reviewer 3 Report
Comments to the Author Manuscript Membranes -1555227
The manuscript entitled ‘Formation of nanowires of various types in the process of galvanic deposition of iron group metals into the pores of track membrane’ focuses on the deposition of mono- and bimetallic nanowires in the pore of the track etched membranes using electrochemical template deposition technique. This study is not absolutely novel and a series of similar experiments were previously published by this group of authors, i.e. [1–3] and others [4, 5]. The presence of similar studies that are quite close in terms of topics as well as low validity of the novelty and significance of this work in the introduction section are significantly reduced the scientific value and novelty of proposed manuscript.
However, after reading this manuscript, there are a lot of points which must be edited or clarified by providing additional information or comments. Publication of the manuscript in this form is impossible.
- Abstract should be elaborated and concretized. Authors stated in lines 22-23 “For the studied types of nanowires, a relationship was established between the growth conditions and the structure and magnetic properties” but in the main text of manuscript there are no any results on magnetic properties et al. This sentence should be concretized.
- In the introduction section, authors should improve/elaborate the goal and novelty of the research. The length of introduction section has to be decreased. In this form, there are many well-known facts and statements in the introduction (lines 1-80). The authors need to conduct a critical analysis of previously published works, thereby substantiating the novelty of this study. The purpose of the research should also be clearly stated.
- The authors should write the complete terms of all abbreviations (including the instruments) before the first use in the abstract and main manuscript i.e. HAADF, butanediol et al.
- The authors have to clarify: the X-ray diffraction patterns (Figure 1b) belong to the arrays of nanowires after removal of the polymer template. How you can explain such an amorphous character of the spectra and, consequently, a low degree of crystallinity of nanowires, and why such a small range of 2Θ? The authors should add a discussion of these spectra and explain their difference? The manuscript as a whole has poor discussion of the actual findings.
- Line 315-317: The authors state that the composition of the electrolyte changed. Please specify how it was performed?
- Line 333: "the ratio of the concentrations of metals in iron-nickel NW differs markedly from the ratio in the growth electrolyte" please explain this statement. How concentrations of nanowire components were determined? As a whole, the paragraph (lines 333-342) needs to be carefully elaborated. Again how increasing of concentration was determined?
- A very important point: why are there no any figures related to the Iron-Nickel nanowires, only a brief discussion in the text? This critical moment must be resolved. It is also recommended to add several SEM images of multicomponent nanowires.
- Please provide details for “standard” deposition conditions and how it was “modified” (lines 377-379)
- Again, in section 3.2.2 authors declaimed about Co/Cu and Ni/Cu nanowires but all graphical data as well as discussion were only about Ni/Cu NWs. Please provide missing data.
- Please clarify about crystallinity degree of multicomponent NWs and provide XRD and SEM data.
- The conclusion section should be elaborated and improved. The author should bring specific conclusions in accordance with obtained results.
- I recommend verifying the English scientific writing, some sections need improvement. Please check the whole manuscript against typos.
Our decision on this manuscript – Major revision. Only after making all reqsubstantial changes in article it could be recommended for publication.
- Zagorskiy DL, Doludenko IM, Kanevsky VM, et al (2021) The Obtaining, Microscopy, and Properties of FeCo and FeNi Alloy Nanowires. Bull Russ Acad Sci Phys 85:848–853. https://doi.org/10.3103/S106287382108030X
- Zhigalina OM, Khmelenin DN, Ivanov IM, et al (2021) Structure of the Fe–Co Nanowires Obtained by Template Synthesis. Crystallogr Reports 66:1109–1116. https://doi.org/10.1134/S1063774521040246
- Panina L V., Zagorskiy DL, Shymskaya A, et al (2021) 1D Nanomaterials in Fe‐Group Metals Obtained by Synthesis in the Pores of Polymer Templates: Correlation of Structure, Magnetic, and Transport Properties. Phys Status Solidi 2100538:2100538. https://doi.org/10.1002/pssa.202100538
- Salman A, Sharif R, Javed K, et al (2020) Controlled electrochemical synthesis and magnetic characterization of permalloy nanotubes. J Alloys Compd 836:155434. https://doi.org/10.1016/j.jallcom.2020.155434
- Mansouri N, Benbrahim-Cherief N, Chainet E, et al (2020) Electrodeposition of equiatomic FeNi and FeCo nanowires: Structural and magnetic properties. J Magn Magn Mater 493:165746. https://doi.org/10.1016/j.jmmm.2019.165746
Author Response
Dear Reviwer!
Thank you for very careful and detailed study of our text! Your suggestions are very important and contribute to the improvement of the text.
We make corrections according to your recommendations-Please, see the attachment.
Sincerely yours- Dmitry Zagorskiy

Round 2
Reviewer 3 Report
No doubt, in revision form this manuscript looks much better. The authors took into account almost all the suggestions of the reviewers and made appropriate corrections to the text of the manuscript.
I have one unresolved request - I keep insisting that the introduction is too long, too bored and needs to be extensively revised. Authors should specify this section of the manuscript and reduce it.